# Isolation, Identification and Antibacterial Characteristics of *Lacticaseibacillus rhamnosus* YT

**DOI:** 10.3390/foods13172706

**Published:** 2024-08-27

**Authors:** Chengran Guan, Feng Li, Peng Yu, Xuan Chen, Yongqi Yin, Dawei Chen, Ruixia Gu, Chenchen Zhang, Bo Pang

**Affiliations:** 1Key Lab of Dairy Biotechnology and Safety Control, College of Food Science and Engineering, Yangzhou University, Yangzhou 225127, China; crguan@yzu.edu.cn (C.G.); 13132197079@163.com (F.L.); yupengjsha@163.com (P.Y.); chenxuan961218@126.com (X.C.); dwchen@yzu.edu.cn (D.C.); grls2008@126.com (R.G.); cczhang@yzu.edu.cn (C.Z.); 2College of Food Science and Engineering, Yangzhou University, Yangzhou 225127, China; yqyin@yzu.edu.cn

**Keywords:** *Lacticaseibacillus rhamnosus* YT, fermented food safety, biopreservation, antibacterial characteristic

## Abstract

Pathogenic microorganisms have been detected in fermented food. Combining the enormous class of the pathogens and their continuously appearing mutants or novel species, it is important to select suitable and safe antibacterial agents for fermented food safety. Lactic acid bacteria (LAB) which produce diverse imperative antimicrobial metabolites have an immense number of applications in the food industry. Here, the human-derived strain YT was isolated due to its cell-free supernatant (CFS-YT) and cells (Cs-YT), respectively performed obvious inhibitory ring to Gram-positive and -negative spoilage bacteria. Strain YT was identified as *Lacticaseibacillus rhamnosus* by the 16s rDNA sequence and morphology. The antibacterial activity of CFS-YT was demonstrated to be growth-dependent, pHs-sensitive, broadly thermostable and enzyme-insensitive. Cs-YT displayed a broad antibacterial spectrum with the action mode of bacteriostasis. The antibacterial activity of Cs-YT was due to substances located at the cell surface which were sensitive to heat, stable at broad pH gradients and sensitive to specific enzymes. These data suggested that *L. rhamnosus* YT could be used as an alternative antimicrobial agent in fermented food biopreservation.

## 1. Introduction

Fermentation is a valid and successful method for food to extend the shelf-life and improve organoleptic properties. Fermented foods mainly produced by natural or well-defined processes have been historically one of the most important parts of the human diet [1]. However, during food production, the presence of pathogenic microorganisms or toxic agents contributes to cases of illness or even outbreaks [2]. Pathogenic microorganisms such as *Escherichia coli*, *Salmonella* spp., *Staphylococcus aureus* and *Listeria monocytogenes* have been detected in fermented food [3]. In many countries, foodborne diseases caused by consuming fermented foods contaminated by pathogenic microorganisms have been reported each year [4]. Foodborne diseases caused by microorganism are the most prominent problem in fermented food quality and safety. Maintaining microbiological fermented food safety is an important task for all countries to ensure national security. Currently, as consumers are more and more health conscious, safe bacteriostatic agents without toxic side-effects are becoming attractive.

Lactic acid bacteria (LAB) has been widely used in the food and pharmaceuticals industries because of its probiotic function as well as the status of GRAS (generally regarded as safe) and QPS (qualified presumption of safety). So far, LAB has been shown to be an effective strain with antibacterial activity for inhibiting foodborne pathogenic bacteria in numerous studies. LAB can synthesize a variety of metabolites with antibacterial activity, such as hydrogen peroxide, fatty acids, organic acids, ethanol, lytic enzymes and bacteriocin [5]. Antibacterial agents may differ in many respects, including the antibacterial spectrum, biosynthetic mechanisms, structural characteristics, modes of antimicrobial action, sources of isolation and specific activity. For example, bacteriocins were different in the antibacterial spectrum, plantaricin JLA-9 showed broad-spectrum antibacterial activity against Gram-positive and -negative bacteria, especially *Bacillus* spp. [6], bacteriocins ST28MS and ST26MS displayed inhibitory activity against the Gram-negative bacteria of *Acinetobacter*, *Escherichia* and *Pseudomonas* and bacteriocins JW3BZ and JW6BZ demonstrated to be active against a wide range of Gram-positive bacteria [7,8]. Moreover, bacteriocin produced by *Lysinibacillus* JX402121 was found highly active against *B. pumilus*. However, given that 14 novel species of *Lysinibacillus* have been identified so far, none have been able to act against *B. pumilus* [9]. 

The identified antibacterial metabolites of LAB were mainly isolated from the cell-free supernatant. Recently, a few studies have found that the antibacterial activities of LAB were related to the active substances on the surface of the bacteria including surface proteins, polysaccharides, teichoic acid, etc. Cytoplasmic hydrolases isolated from *Lactobacillus acidophilus* displayed an inhibitory effect on the *E. coli* growth and the extracellular polysaccharide of *Lactiplantibacillus plantarum* YW32 was able to inhibit the biofilm formation of *E. coli*, *S. aureus*, *Shigella fowleri* and *Salmonella typhimurium* [10,11]. Furthermore, there were LAB in which the cell-free supernatant and the cell pellets did not simultaneously possess the antibacterial activity. Yang et al. [12] observed that the CFS of *Limosilactobacillus reuteri* AN417 performed antibacterial activity against *Porphyromonas gingivalis*, *Fusobacterium nucleatum* and *Streptococcus mutans*. However, the cell extracts of *L. reuteri* AN417 had no antimicrobial activity towards the three oral pathogens.

Combining the enormous class of the pathogens and their continuously appearing mutants or novel species, it is important to select the suitable agents with antibacterial activity for food safety. Therefore, using the common foodborne pathogens and spoilage bacteria as indicator strains, the aim of this study was to screen LAB with an excellent inhibiting ability from human-derived LAB as they are safe to human health and they have already adapted to the human environment. This work would be helpful to develop natural preservatives for microbiologically safe fermented products.

## 2. Materials and Methods

### 2.1. Strains and Growth Conditions

The indicator strains used in this study were purchased from the China Center of Industrial Culture Collection (Beijing, China), including *Bacillus subtilis* CICC10012 (*B. subtilis*), *Salmonella enterica* WX29 (*S. enterica*), *Staphyloccocus aureus* CICC10201 (*S. aureus*), *Bacillus cereus* ATCC11778 (*B. cereus*), *Escherichia coli* CICC10899 (*E. coli*) and *Pseudomonas brenneri* CICC10271 (*P. brenneri*). All the strains were cultivated in Luria Bertani (LB) broth with aeration at 37 °C. The LAB were preserved by the Key Laboratory of Dairy Biotechnology and Safety Control of Jiangsu Province which were isolated from the gut of long-lived population from Bama, Guangxi province, China. All the LAB strains were inoculated in deMan, Rogosa and Sharpe (MRS) broth at 37 °C in static condition.

### 2.2. Measurement of Antibacterial Activity

Antibacterial activity was determined by agar well diffusion method. Briefly, the colony of the indicator bacteria was inoculated into a tube containing 5 mL of LB medium and cultivated at 37 °C for 12 h. The agar plate was pre-prepared by pouring 30 mL of LB agar medium into a plate with 90 mm diameter. And then 100 μL of indicator bacteria suspension diluting to 1.0 × 10^6^ CFU/mL was used to spread on the plate. After drying for 2 h, 7 mm-diameter wells were made in the plate using a sterile punch. Two hundred microliter of the tested sample solution was added into the well and spread at 4 °C for 4 h. Then the plate was transferred to an incubator at 37 °C for 8 h. Antimicrobial activity was determined by measuring the diameter size of the clear zone around the wells (excluding the 7 mm hole).

### 2.3. Screening of LAB with Antibacterial Activity

To resuscitate the strains, the colony of the LAB was inoculated into MRS broth and continuously cultivated three times at 37 °C for 18h. Thereafter, the LAB culture was incubated into 10 mL of MRS broth at 37 °C for 24 h and then used for assay of antibacterial activity. Firstly, the cultures were gently mixed and then the antibacterial activity to the six indicator strains was measured. According to the diameter of the inhibition zone, the LAB strains were sorted into 5 groups: “-”, “+”, “++”, “+++”, “++++” indicating that the diameter of the inhibition zone was zero, equal or lesser than 3.00 mm, from 3.01 mm to 6.00 mm, from 6.01 mm to 10.00 mm and above 10.01 mm, respectively. The strains with a diameter of the inhibitory zone which was higher than zero were selected. Secondly, the culture of the screened strains was centrifuged at 8000 rpm/min for 10 min at 4 °C and the cells and the supernatant were individually collected. The pH of the supernatant was adjusted to 5.0 (±0.05) with 3 M NaOH solution and used to re-screen samples possessing antibacterial activity against both *B. subtilis* and *S. enterica*. Lastly, the cell pellets of the re-screened sample were washed twice with deionized water (ddH_2_O), then re-suspended in an equal volume of ddH_2_O. The cell suspension showing antibacterial activity to both *B. subtilis* and *S. enterica* was selected and the corresponding strain was used for further work.

### 2.4. Molecular Identification

The chromosome DNA of the isolated strain was firstly extracted using an Ezup column bacterial genomic DNA extract kit (Sangon, Shanghai, China). The 16s rDNA was amplified from the corresponding chromosome DNA with universal primers 16s-27F (5′-AGAGTTTGATCCTGGCTCAG-3′) and 16s-1492R (5′-TACGGCTACCTTGT TACGACTT-3′). The PCR product was purified and sent to be sequenced (Bioengineering Co., Ltd., Shanghai, China). The 16s rDNA sequence was analyzed by the NCBI online blast. The strain was named as *Lacticaseibacillus rhamnosus* YT and its 16s rDNA sequences were deposited in GenBank with the submission number of ON705254.

### 2.5. Growth and Antibacterial Curve of the Supernatant

Nine milliliter culture of *L. rhamnosus* YT was inoculated into 300 mL of MRS broth and cultivated at 37 °C for 24 h during which the cultivation was sampled at 3 h intervals. The sample was centrifuged (10,000 rpm, 4 °C, 10 min) to obtain the cell pellets and the supernatant. The cell pellets were washed twice by PBS buffer (10 mM, pH 7.0) and resuspended with the same volume of PBS buffer. The antibacterial activity of the supernatant and the pellets suspension was measured by agar well diffusion method. Growth of *L. rhamnosus* YT was monitored by measuring the optical density of the pellets suspension at 590 nm with a 1510-spectrophotometer (Thermo Fisher Scientific Oy, Vantaa, Finland).

### 2.6. Viable Count Method

The viable bacterial counts were performed as described [13]. Briefly, the sampled solution was centrifuged (10,000 rpm, 4 °C, 10 min), obtaining the cell pellets. After washing twice with saline solution, the cell pellets were resuspended with the same volume of saline solution. Then, 1 mL of the sample was diluted with 9 mL of saline solution and eight serial dilutions were performed. Each bacterium was counted in the three most appropriate dilutions by applying the pour plate technique.

### 2.7. Co-Incubation with Indicator Bacteria Assay

The indicator bacteria *B. subtilis* and *S. enterica* were separately made by cultivation into a tube containing 5 mL of LB medium at 37 °C for 12 h. The strain YT solution was prepared by suspending the cell pellets of *L. rhamnosus* YT in the PBS buffer (10 mM, pH 7.0) at 1 × 10^8^ CFU/mL. Using the same volume of PBS buffer as control, 1 mL of the strain YT solution were added into LB broth containing 1% (*v*/*v*) of freshly prepared indicator bacteria. The co-incubation was processed in a shaker at 37 °C for 10 h, during which the culture medium was sampled every 2 h. The viable bacterial counts of the indicator bacteria in every sample were counted.

### 2.8. Sensitivity of Antimicrobial Activity

*L. rhamnosus* YT was inoculated with MRS broth and cultivated at 37 °C for 24 h. The cell-free supernatant and cell pellets suspended in the PBS buffer were obtained using the usual procedure and were respectively used to detect the sensitivity of antimicrobial activity. To test the sensitivity to pH, the sample was adjusted to different pHs in the range of 3.0–9.0 with HCl (4 M) or NaOH (4 M). To evaluate the heat stability, the sample was treated at 30 °C, 40 °C, 50 °C, 60 °C, 70 °C, 80 °C, 90 °C and 100 °C for 15 min, respectively. To test the impact of enzymes, pepsin, trypsin, papain, protease K, catalase, α-amylase, β-amylase, β-glucosidase and neutral protease (Sinopharm Chemical Reagent Co., Ltd., Shanghai, China) were used. Every enzyme (50 mg/mL) was prepared in advance. The enzyme solution was added to the tested sample to the concentration of 2 mg/mL at the corresponding optimal reaction pH at 37 °C for 2 h. MRS broth and PBS buffer with the same treatment were used as the control of the cell-free supernatant and the cells’ solution, respectively. The antimicrobial activity of the treated sample was measured by agar well diffusion method. The viable counts of *L. rhamnosus* YT were detected as previously described.

### 2.9. Statistical Analysis

Statistical analysis was carried out by SPSS 19.0 software (SPSS Inc., Chicago, IL, USA). Data of 3 independent experiments were statistically analyzed by one-way analysis of variance (ANOVA) and expressed as mean ± standard deviation (SD).

## 3. Results and Discussion

### 3.1. Screening and Identification of LAB Strain with Antibacterial Ability

In this work, the Gram-positive (*B. subtilis*, *S. aureus* and *B. cereus*) and Gram-negative (*E. coli*, *S. enterica* and *P. brenneri*) bacteria, which were generally found in the polluted foods, were used as indicator strains for isolated human-gut-derived LAB with antibacterial activity. The concentration of the indicators used for spreading on the plate was optimized as 1 × 10^6^ CFU/mL according to the appearance of the layer formed by the indicator strain and the inhibitory ring. Firstly, the antibacterial activity of the 102 LAB fermentation cultures was measured separately. As shown in Table 1, the LAB showed various antibacterial activities against the indicator strains and 65 strains were selected which displayed an obvious antibacterial zone to all the 6 indicator strains. Afterwards, to simplify the experiment, *B. subtilis* and *S. enterica* were used as the indicator strains in the followed work. The pH of the cell-free supernatant (CFS) of the 65 LAB was individually regulated to 5.0, and only 6 strains represented a distinct antibacterial zone for *B. subtilis* and *S. enterica* (Table 2). Finally, the cell pellets of the six strains were resuspended in ddH_2_O and only strain YT cells displayed significant inhibitory activity in both indicator bacteria (Table 3). During the selection process, the inhibitory activity of *S. enterica* in some LAB strains were used as representatives, as shown in Figure 1a,b.

The colony of strain YT displayed typical morphological characteristics of LAB with a rod shape, white color, translucent opacity, smooth and creamy surface (Figure 1c) and Gram-positive staining (Figure 1d). The 16s rDNA was cloned (Figure 1e) and the sequence was aligned with the deposited sequences in GenBank. Strain YT possessed a 16s rDNA sequence with 98.62% similarity to *Lacticaseibacillus rhamnosus* strain 6655. Combined with the morphological and physiological properties, strain YT was identified and named as *L. rhamnosus* YT and then stored in the China General Microbiological Culture Collection Center with the corresponding No. 22693.

### 3.2. Antibacterial Characteristics of the CFS from L. rhamnosus YT (CFS-YT)

#### 3.2.1. Growth Kinetics and Antimicrobial Activity of the CFS-YT

To explore the relationship between the antibacterial activity of the CFS-YT and the strain growth, *L. rhamnosus* YT was cultivated for 24 h and the CFS-YT was prepared at distinct time points. *L. rhamnosus* YT went into a stationary phase after 18 h in which the biomass was 1.75 times higher than the initial inoculation amount (Figure 2). The pH of the CFS-YT was continually decreased and the corresponding antibacterial activity was constantly increased during the logarithmic growth phase. After reaching the stationary phase, the pH and the antibacterial activity of CFS-YT almost remained stable. Some of the antibacterial substances discovered in the supernatant from LAB were also mainly accumulated during the exponential growth phase [14]. Milioni et al. [15] found that the maximum bacteriocin was obtained when *L. plantarum* LpU4 entered the late exponential phase (24 h) and remained consistently during the stationary phase up to 48 h.

Moreover, CFS-YT showed higher antibacterial activity for *S. enterica* than for *B. subtilis.* The biggest inhibitory zone for *B. subtilis* and *S. enterica* were 12.17 mm at 21 h and 15.50 mm at 18 h, respectively. The highest antibacterial activity and the parallel culture time were different against the two indicator strains. These data indicated that there were different antibacterial substances in CFS-YT aiming at specific indicator strains.

#### 3.2.2. The Characteristics of the Antimicrobial Activity of CFS-YT

The antimicrobial activity of CFS-YT appeared to be maximal at acidic pHs and decreased at more neutral and alkaline pHs (Table 4). When tested at pHs higher than or equal to 6.0, the antimicrobial activity of CFS-YT was no longer presented. It was reported that the antibacterial substances sensitive to pH were organic acids (lactic acid, acetic acid, phenyl lactic acid, etc.), bacteriocin and peptides [16]. Hussain et al. [17] found that acid neutralization but not protease treatment abolished the antibacterial activity of the CFSs of four LAB isolates and the antibacterial components in the cell-free supernatants were lactic, acetic and propionic acids. Plantaricin ZJ008 produced by *L. plantarum* ZJ008 exhibited narrow pH stability (pH 4.0–5.0) and plantaricin JY22 from *L. plantarum* JY22 was activated at pHs ranging from 2.5 to 5.5 [18,19]. Moreover, the antibacterial activity of *L. johnsonii* La1 and *L. plantarum* ACA-DC 287 was found to be caused by the production of lactic acid and (an) unknown inhibitory substance(s) which was (were) only active in the presence of lactic acid [20]. Therefore, it was necessary to explore the factors affecting the antibacterial activity of CFS-YT which were helpful to identify the antibacterial substances.

To determine the heat sensitivity, CFS-YT was disposed at different temperatures. The antibacterial activity of CFS-YT was heat stable (Figure 3a). Even after treating at 100 °C for 15 min, the antibacterial activity of CFS-YT against *B. subtilis* and *S. enterica* was merely reduced by 4.56% and 4.01%, respectively. These data suggested that most of the antibacterial substances in CFS-YT were heat stable. Moreover, the slight decline in antibacterial activity at 100 °C indicated that there might be scarce heat-sensitive materials. Until now, the previously studied antibacterial substances from the CFS were distinct in their heat sensitivity. IUPAC, coagulin A, bacteriocin HD1-7, plantaricin ZJ008 and bacteriocin PFC69 remained stable at temperatures higher than 100 °C [18,21,22,23,24]. However, antibiotic AS-48 and bacteriocins BLh were sensitive to heat at temperatures varied from 60 °C to 80 °C [25,26]. It was presumed that heat affected the antibacterial materials by destroying their advanced structure or making the thermal instability materials decompose or volatilize.

To determine the enzymes’ sensitivity, CFS-YT was treated by different enzymes under respective optimum conditions. The antibacterial activity of CFS-YT remained consistent after treatment with pepsin, papain, trypsin and proteinase K (Figure 3b). Danilova et al. [27] found that the supernatant of *L. plantarum* retained bactericidal activity after treatment with proteinase K and supposed that the antimicrobial activity of the supernatant was not associated with the protein or peptide fraction. Therefore, the antimicrobial activity of CFS-YT against *B. subtilis* and *S. enterica* might not be due to the protein antibacterial substance or the protein antibacterial substances may not have the main effective substances for antibacterial activity. By cultivating with catalase, the antibacterial zones of CFS-YT against *B. subtilis* and *S. enterica* were, respectively, reduced by 10.21% and 16.24%, suggesting that there was some hydrogen peroxide in CFS-YT. Liu et al. [28] found that there was no significant difference in the diameter of the inhibition zone of the three LAB fermentation broths after catalase treatment, indicating that hydrogen peroxide was not the main antibacterial active substance.

In summary, under the conditions tested, the antimicrobial activity of CFS-YT was due to (a) heat-stable, acidic-pH-active and enzyme-insensitive compound(s), indicating that the antibacterial substances in CFS-YT were metabolites like organic acids, hydrogen peroxide and/or nonproteinaceous low-molecular-weight compound(s). The composition and content of the specific antibacterial components require further research. Our results demonstrate that CFS-YT had antimicrobial activity against broad foodborne bacteria and could be applied to microbiological control in food products, especially the production and storage conditions which were acidic at broad temperatures.

### 3.3. Antibacterial Characteristics of the L. rhamnosus YT Cells (Cs-YT)

#### 3.3.1. Antibacterial Activity of Cs-YT

To date, the antibacterial agents found in LAB were mainly isolated from the supernatant. There was relatively little work about cell pellets with antibacterial activity. Here, Cs-YT displayed a wide antibacterial spectrum against both Gram-positive and -negative indicator strains (Table 5). In addition, Cs-YT exhibited higher antibacterial activity against the Gram-positive strains than the Gram-negative strains which might because Gram-positive stains had thick peptidoglycan cell walls. These results suggested that Cs-YT is an attractive candidate to explore novel antibacterial agents and is used to control foodborne pathogens in fermented food.

#### 3.3.2. Mode of Action

To date, there have been scarce studies about active substances on the cell surface with an antibacterial mode of action. In this work, after individually co-incubated with Cs-YT, both *B. subtilis* and *S. enterica* exhibited growth suppression and their biomass was obviously reduced in the stationary phase (Figure 4). Moreover, the lag phase of the indicator strains was obviously delayed after co-incubation. These results suggested that the action mode of Cs-YT was bacteriostasis. Antimicrobial materials have the ability of inhibiting the growth or even killing certain types of microorganisms. The main antibacterial substances of CFS from *L. plantarum* DY-6 were lactic acid, acetic acid, propionic acid, caprylic acid and decyl acid and the bacteriostatic action destroyed the stability of the cell membrane of the pathogenic bacteria [29]. In addition, the CFS of *L. crispatus* was capable of killing harmful pathogens [30].

#### 3.3.3. Factors Affecting the Activity of Cs-YT

Cs-YT suspended in the PBS buffer was treated at different temperatures, pHs and enzymes (Figure 5). The viable counts of Cs-YT remained consistent with the antibacterial activity during these treatment which suggested that the antibacterial activity was caused by the substances located on the cell surface instead of the intracellular materials. The antibacterial activity of Cs-YT was stable at 30 °C and 40 °C and then reduced along with the increased temperature. There were no viable bacteria or antibacterial ring at temperatures higher than or equal to 70 °C. Although the viable counts at 50 °C remained stable, the antibacterial activity of Cs-YT was obviously decreased. These results indicated that the antibacterial activity of Cs-YT was probably caused by the heat-sensitive substances located on the cell surface. After Cs-YT was treated with a series of pH values, the living cells were scarcely influenced and the antibacterial activity against the two indicator strains had slight variations (Figure 5b). In addition, the bacteriostatic effect and the viable cells of Cs-YT were not significantly influenced by the addition of enzymes. Unlike the effects caused by heat and pHs, Cs-YT treated by enzymes performed different variation trends of inhibitory zones between the two indicator strains (Figure 5c). The antibacterial activity of Cs-YT against *B. subtilis* and *S. enterica* was, respectively, reduced by the treatment of α-amylase and β-amylase. The antibacterial activity of Cs-YT against *B. subtilis* and *S. enterica* was increased by the management of β-glucosidase and pepsin, respectively. In addition, the antibacterial activity was stable with the other enzymes. Therefore, the antibacterial materials from the cell surface of *L. rhamnosus* YT were probably composed of more than one element, including proteins, peptides and carbohydrates.

According to the variation of the viable counts, the intracellular substances were not involved and the antibacterial substances were cell bound. The bioactive substances that have been reported on the surface of LAB cells are mainly teichoic acid, surfactants, surface proteins, extracellular polysaccharides and phosphopeptides [10,31,32,33]. Bacterial lipoteichoic acid (LTA) is an amphiphilic molecule containing glycerolphosphate or ribitolphosphate residues. LTA of *L. plantarum* can inhibit the biofilm formation of pathogenic bacteria, such as *S. mutans*, *E. faecalis* or *S. aureus* [34]. Surface-layer proteins (SLPs) are crystalline arrays composed of a single protein or glycoprotein. SLPs from *L. acidophilus* were found acting against the cell wall of *S. enterica* [35]. Biosurfactants (BSs) are surface-active compounds including types of lipopeptides, glycolipids, glycopeptides and glycolipoproteins. Recently, some papers have described BSs produced by LAB as antimicrobial [36]. In this work, the antibacterial substances on the surface of Cs-YT were presumably a kind of BS by comparing the action mode and physiochemical characteristics of the reported substances and Cs-YT. The antimicrobial substances and properties varied widely among the strains. Hence, the Cs-YT-bound antibacterial components will be purified and identified in our subsequent work.

## 4. Conclusions

*L. rhamnosus* YT was isolated and identified by the morphology and 16s rDNA sequence. Both CFS-YT and Cs-YT of *L. rhamnosus* YT performed obviously inhibitory activity against Gram-positive and -negative spoilage bacteria. The antibacterial activity of CFS-YT increased along with the cell growth and was pH-sensitive, heat-stable and enzyme-insensitive. Cs-YT was a bacteriostatic mode of action and the corresponding antibacterial substances were cell-bound, and so were sensitive to heat, stable at broad pH gradients and sensitive to specific enzymes. These results suggested that *L. rhamnosus* YT would be an attractive candidate for antibacterial agents applied in the fermented food industry.

## Figures and Tables

**Figure 1 foods-13-02706-f001:**
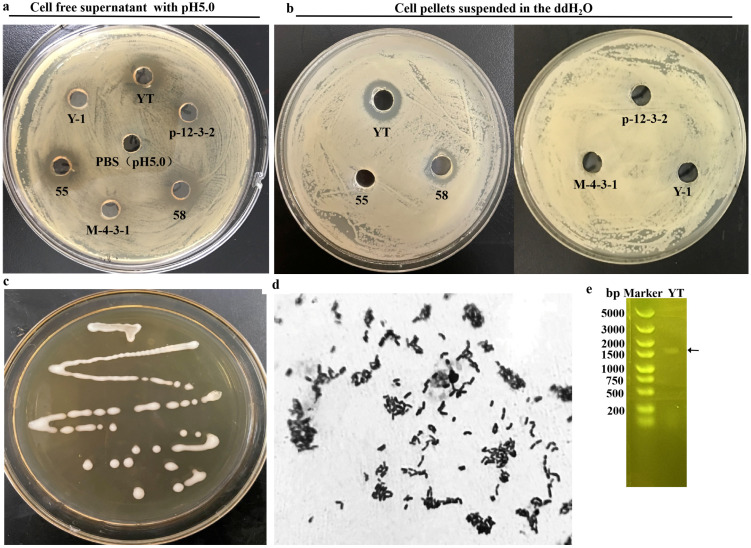
Re-screening of LAB strain with antibacterial activity by agar well diffusion method. The inhibition zone of the supernatant with pH 5.0 against *S. enterica* (**a**) and the cell pellets suspended in ddH_2_O (**b**). The colony of strain YT (**c**) and its cell morphology observed by Gram staining (**d**). Agarose gel electrophoresis of 16s rDNA band amplified from strain YT (**e**), arrow indicating the target band.

**Figure 2 foods-13-02706-f002:**
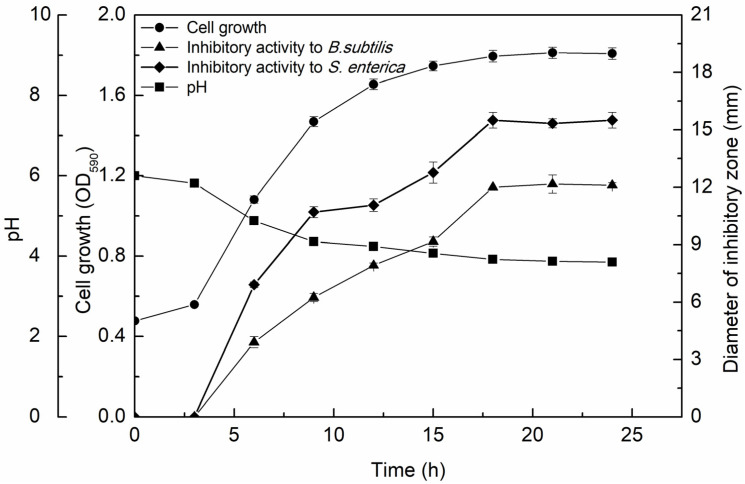
Growth kinetics and antimicrobial activity of the CFS-YT.

**Figure 3 foods-13-02706-f003:**
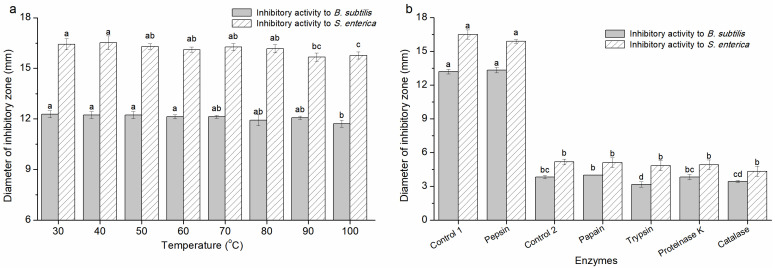
The sensitivity of the antimicrobial activity of CFS-YT to heat (**a**) and enzymes (**b**). Control 1 meant control to pepsin (PBS with pH 3.8) and control 2 meant control to papain, trypsin, proteinase K and catalase (PBS with pH 5.0). Different letters indicating significant differences (*p* < 0.05).

**Figure 4 foods-13-02706-f004:**
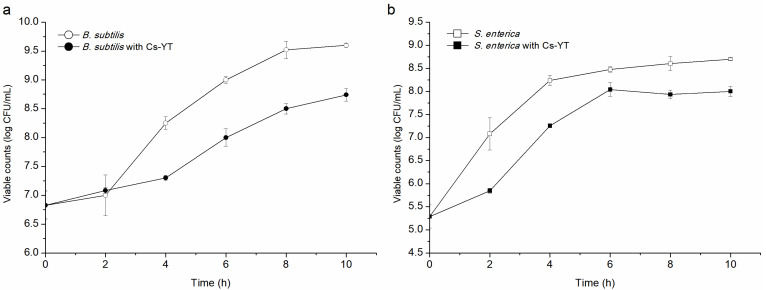
The cell growth of *B.subtilis* (**a**) and *S. enterica* (**b**), respectively co-cultivated with Cs-YT.

**Figure 5 foods-13-02706-f005:**
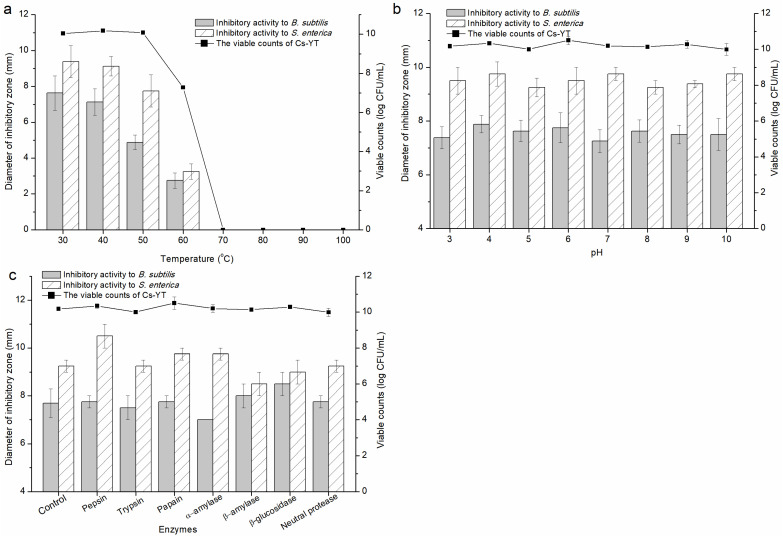
The sensitivity of the antimicrobial activity of Cs-YT to heat (**a**), pH (**b**) and enzymes (**c**).

**Table 1 foods-13-02706-t001:** Initial screening results of LAB with antibacterial activity.

Strain	Inhibitory Zone against Indicator Strains (mm)					
	*B. subtilis*	*S. aureus*	*B. cereus*	*S. enterica*	*E. coli*	*P. brenneri*
2	++	++	+	+	++	++
4	++	+	-	+++	+++	+++
6	+	++	-	+	+	+++
7	++	+++	+++	+++	+++	+++
9	+	++	+	+	+++	+++
10	+	+	+	+	++	++
11	++	+++	+	+	+++	+++
18	-	-	-	-	-	-
21	++	++	+++	++	++	++
25	+	++	+	+	++	++
27	+	+	+	+	++	+++
31	+	+	+	++	++	++
32	-	-	-	-	-	-
33	+	+	-	+	++	++
34	-	-	-	-	-	-
35	+	+	+	+	++	++
44	++	++	++	+	++	++
46	+	++	+	+	++	++
53	-	-	-	-	-	-
55	++++	++++	+++	+++	++++	++++
57	-	++	+	+	+++	++
58	++++	++++	++++	++++	++++	++++
61	-	-	-	-	-	+
65	+	+	-	-	++	+++
66	+	++	+	+	++	++
69	++	+	++	++	++	+++
71	+	+	+	+	+++	++
74	++	+	+++	+	+++	++
79	++	++	++	-	+++	+++
82	-	++	+	+	+++	++++
85	+	++	++	+	++	+++
87	++	++	+	-	+++	+++
88	+	-	+	+	++	++
89	+	+	+	+	++	+++
96	+	+++	++	+	++	++++
97	++	++	++	+	+++	+++
98	-	-	-	-	-	-
99	-	+	+	++	++	+++
103	+	+	+	+	+	+
105	+	+	+	++	++	+++
106	++	++	++	+	+++	+++
111	++	++	++	+	++	++
113	+++	++	++	+	++++	++++
116	++	++	++	+	+	++
117	+++	+++	+	+	+++	+++
118	++	++	-	++	++	+++
145	++	++	++	-	++	++
146	+	+	+	++	++	++
147	++	+	+	+	+++	+++
148	+	+	+	+	++	++
154	++	+++	++	+	+++	++
393	+	+	+	+	++	+
p-1-3-2	++	-	-	+	++	++
p-1-3-9	+++	++	++	++	++++	++
p-1-7-4	+++	++	++	++	++++	++
p-3-3-6	++	++	+	++	+++	++
p-11-7-6	-	-	-	-	-	-
p-12-3-2	++++	+++	++++	++++	++++	++++
p-13-7-11	++	+++	++	++	+++	++
p-14-3-2	+	++	++	++	+++	+++
p-14-7-3	-	+	+	++	++	+++
M-4-3-1	++++	++++	++++	++++	+++	++++
M-14-3-1	++	++	++	+	+	++
M17-6-3-2	+++	+++	++	+	++	++
S-1-3-1	++	+++	++	+	+++	++
S-3-3-1	+	++	+++	-	-	++
S-5-3-2	++	++	++	+	++	++
S-11-3-1	++	+	-	++	++	++
S-13-7-2	++	-	++	++	++	++
S-14-7-5	++	++	++	+	++	+++
L-4-3-2	++	+	-	++	++	+++
L-11-7-2	++	-	+	++	++	+++
L-13-7-7	++	+	+	++	++	+++
L-14-7-3	++	+	+	++	++	+++
Y-1	++++	++++	+++	++++	++++	++++
Y-2	++	++	-	++	++	+++
Y-3	++	+	+	++	++	++
Y-4	+	+	+	+	++	++++
Y-5	+++	++	++	+++	++++	++++
YT	++++	++++	++++	++++	++++	++++
Grx10	++	++	++	+	++	++
Grx19	++	+	+	++	+++	+++
V9	++	++	++	++	-	+
LH100-1	+	+	-	+	+++	+++
LV108	+++	+	+	++	++	++
R3	+++	-	+	+	+	+
90-57	++	+	+	++	+++	+++
St KY	+	+	+	+	++	++
S7	++	+	-	+	++	-
087-3	+	+	+	++	++	++
1.12.8	++	+	-	++	++	++
987-3	-	-	-	-	-	-
937-MQ	+	++	++	++	+++	+++
975-1	+	++	++	++	++	++
XPL	+	+	+	++	++	++
XPL-1-4	-	-	-	-	-	-
K01-4	++	++	++	+++	+++	+++
JXR	+++	-	+	+	+	+++
12-529	+	+	-	+	++	++
LD	+++	++	+	++	++	+++
5-7-7-2	++	++	+	+	++	++
9-10-1-10	+	-	+	++	++	+

“-” indicating no inhibition zone; “+” indicating the diameter of inhibition zone equal or lesser than 3.00 mm; “++” indicating the diameter of inhibition zone varied from 3.01 mm to 6.00 mm; “+++” indicating the diameter of inhibition zone varied from 6.01 mm to 10.00 mm; “++++” indicating the diameter of inhibition zone above 10.01 mm.

**Table 2 foods-13-02706-t002:** Rescreening of LAB with excellent antibacterial activity.

Strains	Initial pH of CFS	Inhibitory Zone of CFS against Indicator Strains (mm)		Inhibitory Zone of CFS with pH 5.0 against Indicator Strains (mm)	
		*B. subtilis*	*S. enterica*	*B. subtilis*	*S. enterica*
Y-1	4.01 ± 0.01 ^a^	14.50 ± 0.41 ^c^	16.11 ± 0.08 ^e^	8.20 ± 0.41 ^b^	8.89 ± 0.57b ^c^
YT	3.86 ± 0.03 ^c^	19.33 ± 0.47 ^a^	23.50 ± 0.41 ^a^	10.66 ± 0.29 ^a^	13.13 ± 0.19 ^a^
55	3.92 ± 0.02 ^b^	18.73 ± 0.21 ^a^	17.76 ± 0.20 ^d^	10.50 ± 0.41 ^a^	11.50 ± 0.41 ^b^
58	3.83 ± 0.03 ^c^	17.50 ± 0.08 ^b^	21.50 ± 0.41 ^b^	10.47 ± 0.37 ^a^	11.50 ± 0.40 ^b^
M-4-3-1	3.77 ± 0.01 ^d^	19.17 ± 0.24 ^a^	19.33 ± 0.24 ^c^	10.34 ± 0.29 ^a^	9.33 ± 0.24 ^c^
p-12-3-2	3.86 ± 0.03 ^c^	18.50 ± 0.41 ^a^	17.77 ± 0.21 ^d^	10.07 ± 0.09 ^a^	11.33 ± 0.27 ^b^

Different letters indicating significant differences (*p* < 0.05).

**Table 3 foods-13-02706-t003:** The antibacterial activity of the cell pellets suspended in ddH_2_O.

Strain	Inhibitory Zone (mm)	
	*B. subtilis*	*S. enterica*
Y-1	-	-
YT	12.00 ± 0.25 ^a^	12.50 ± 0.25 ^a^
55	-	-
58	-	10.12 ± 0.03 ^b^
M-4-3-1	-	-
p-12-3-2	-	-

“-” indicating no bacterial inhibitory activity; Different letters indicating significant differences (*p* < 0.05).

**Table 4 foods-13-02706-t004:** The effect of pH on the antibacterial activity of CFS.

pH	Diameter of Inhibitory Zone (mm)	
	*B. subtilis*	*S. enterica*
3.0	18.56 ± 0.08 ^a^	22.50 ± 0.41 ^a^
4.0	10.10 ± 0.29 ^b^	12.49 ± 0.15 ^b^
5.0	3.72 ± 0.21 ^c^	5.87 ± 0.27 ^c^
6.0	-	-
7.0	-	-
8.0	-	-
9.0	-	-

“-” indicating no bacterial inhibitory capacity; Different letters indicating significant differences (*p* < 0.05).

**Table 5 foods-13-02706-t005:** The antibacterial spectrum of Cs-YT.

Indicator Strains	Diameter of Inhibitory Zone (mm)
*B. subtilis*	7.63 ± 0.85 ^b^
*S. aureus*	8.00 ± 0.82 ^ab^
*B. cereus*	8.13 ± 1.31 ^ab^
*S. enterica*	9.13 ± 1.65 ^ab^
*E. coli*	8.38 ± 0.95 ^ab^
*P. brenneri*	9.88 ± 1.18 ^a^

Different letters indicating significant differences (*p* < 0.05).

## Data Availability

The original contributions presented in the study are included in the article, further inquiries can be directed to the corresponding author.

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
