# Peer review of "Isolation, Identification and Antibacterial Characteristics of Lacticaseibacillus rhamnosus YT"

_foods, 2024, doi:10.3390/foods13172706_

Round 1
Reviewer 1 Report
Comments and Suggestions for Authors
Manuscript ID: foods-3144880
Title: Isolation and identification of Lacticaseibacillus rhamnosus YT and its antibacterial characteristics
Authors: Chengran Guan, Feng Li, Peng Yu, Xuan Chen, Yongqi Yin, Dawei Chen, Ruixia Gu, Chenchen Zhang, and Bo Pang.
The objective of this investigation was to screen human-derived LAB with excellent inhibiting ability to common foodborne pathogens and spoilage bacteria, with the idea to develop natural preservatives for microbiologically safe fermented products. This study is updated, well done with some potential application. However, there ae some points to call the attention to the authors within the text.
1.The English should be checked throughout the manuscript as for example at some points below
2. In line 11 did you mean “strain” instead of “strian”?. In line 31 did you mean “consuming” instead of “consumping”?. In line 35 did you mean “widely” instead of “wildly”?. In line 62 what do you mean with “literatures”?. In line 87 did you mean “spread” instead of “spreaded”?.
3. In lines 96 and 116, there should be consistency in “rpm”. In Line 114 did you mean ”were inoculated” instead of “was inoculated”?
4. In Line147 “t-test was applied to 147 compare the mean values at a significant level of 5%”, where in the manuscript the t-test was applied?....was it from line 201 onwards???.
5. Line 181 Figure 1d, it is difficult to follow. In Line 188, “Different Letters: indicating significant differences (p<0.05)”, might be better than “Letters: indicating significant differences (p<0.05)”.
6. From Line 259 till line 273, it might be convenient to mention some references of previous works done to sustain the discussion.
Comments on the Quality of English LanguageThe written English should be checked throughout the manuscript at some points
Author Response
We appreciate your valuable comments on our manuscript. These comments and suggestions have been very carefully considered and the manuscript has been revised with modifications, as you kindly suggested.
Based on the order of the comments, the revised points are as listed below.
Comment 1:The English should be checked throughout the manuscript as for example at some points below
Respones 1: The manuscript has been thoroughly checked and modified according to your recommendation.
Comment 2: In line 11 did you mean “strain” instead of “strian”?. In line 31 did you mean “consuming” instead of “consumping”?. In line 35 did you mean “widely” instead of “wildly”?. In line 62 what do you mean with “literatures”?. In line 87 did you mean “spread” instead of “spreaded”?.
Respones 2: We are sorry for the errors. All the spelling mistakes have been modified in the revised manuscript.
Comment 3: In lines 96 and 116, there should be consistency in “rpm”. In Line 114 did you mean ”were inoculated” instead of “was inoculated”?
Respones 3: These mistakes have been corrected as your kind suggestion.
Comment 4: In Line147 “t-test was applied to 147 compare the mean values at a significant level of 5%”, where in the manuscript the t-test was applied?....was it from line 201 onwards???.
Respones 4: The data was analyzed by one-way analysis of variance (ANOVA). The section “2.8. Statistical analysis “ has been revised.
Comment 5: Line 181 Figure 1d, it is difficult to follow. In Line 188, “Different Letters: indicating significant differences (p<0.05)”, might be better than “Letters: indicating significant differences (p<0.05)”.
Respones 5: The title of Figure 1d has been modified as “The colony of strain YT (c) and its cell morphology observed by gram staining (d)”. Besides, “Letters: indicating significant differences (p<0.05)” in the manuscript have been unified as “Different Letters indicating significant differences (p<0.05)” in the revised manuscript.
Comment 6: From Line 259 till line 273, it might be convenient to mention some references of previous works done to sustain the discussion.
Respones 6: Thank you for your recommendation. Two related references have been mentioned in the revised manuscript.
The manuscript had been thoroughly checked and modifications were made. Besides, Figure3, 4 ,and 5 have been redrawn. We really hope the revised manuscript will be acceptable for publication.
Reviewer 2 Report
Comments and Suggestions for Authors
The reviewed manuscript of Chengran Guan et al., addresses the characterization of lactic acid bacteria isolated from human, in particular one isolates identified as Lacticaseibacillus rhamnosus YT could be a beneficial lactic acid bacterium with potential health benefits because showed antibacterial effects. The authors investigate the antibacterial activity of cell-free supernatants (CFS) against Gram positive and Gram negative bacteria, such as Bacillus subtilis, Salmonella enterica, Staphyloccocus aureus, Clostridium difficile and Escherichia coli. On the other hand, with Lacticaseibacillus rhamnosus YT strain some microbiological essays were made with the objective of understand the antibacterial activity, these were pH, temperature and proteases inhibition tested on the CFS. Although the authors present diverse results, I consider that is necessary some major revisions on the manuscript.
Here is my observation:
1. I suggest edit the title for major impact.
2.- In introduction lines 23-28 add some references.
3.-In introduction lines 32-34, I think that is not necessarily this case.
4.- In materials and methods, section 2.1 B. cereus is missing in description of strains as well as Clostridium difficile was not used in the essays. Also, I suggest a more detailed description of LAB used in this work, for example as were isolated from human gut and characterized.
5.- In materials and methods, section 2.2 lines 86-87, What is the reason to spread the sample solution at 4 oC for 4 h?.
6.- In materials and methods, section 2.3 lines 92-93, The three generation as they were determined? What mean bacterial generation?
7.- In materials and methods, section 2.6 lines 128-129, please describe more detailed the count of viable cells.
8.- Section RESULTS, change for Results and discussion
9.- In section of Results lines 151-161 is introduction, please eliminate this text.
10.- In Results section, I suggest change the Table 1 for a figure with graphs or bars to show with clarity the results and a more detailed description of these results, for example which bacterial strains showed better antibacterial activity and selected.
11.- In Results section, Figure 1 and text (lines 165-173), not only the strain YT displayed antibacterial activity, also the isolate 58, Why it was discarded? One point important is that in figure 1 only the results for S. enterica is showed. Please edit the text.
12.- The word “blasted” (line 195) is incorrect, please edit.
13.- In table 2, these values are in mm or what units?
14.- In Results section 3.3, figure 4, The cell growth of B.subtilis and S. enterica co-cultivated with Cs-YT, the results are confusing, because according with methodology is not clear as they were determined, also in these bacterial growth condition could be generate colonization competition, so that from my point of view these essays are not clear for interpretation.
Comments on the Quality of English LanguageMinor editing of english language is required.
Author Response
We appreciate your valuable comments on our manuscript. These comments and suggestions have been very carefully considered and the manuscript has been revised with modifications, as you kindly suggested.
Based on the order of the comments, the revised points are as listed below.
Comment 1: “I suggest edit the title for major impact.”
Response 1: The title has been modified as “Isolation, identification and antibacterial characteristics of Lacticaseibacillus rhamnosus YT” in the revised manuscript.
Comment 2: “ In introduction lines 23-28 add some references.”
Response 2: Thank you for your suggestion. The references have been added in the revised manuscript.
Comment 3: “In introduction lines 32-34, I think that is not necessarily this case.”
Response 3: The sentence has been deleted in the revised manuscript.
Comment 4: “In materials and methods, section 2.1 B. cereus is missing in description of strains as well as Clostridium difficile was not used in the essays. Also, I suggest a more detailed description of LAB used in this work, for example as were isolated from human gut and characterized.”
Response 4: We are sorry for the error that Bacillus cereus was miswritten by Clostridium difficile in section 2.1. Besides, the detailed description of LAB has been supplied in the section 2.1.
Comment 5: “In materials and methods, section 2.2 lines 86-87, What is the reason to spread the sample solution at 4 oC for 4 h?.”
Response 5: Under this condition, the indicator bacteria grow very slowly and would not grow into a layer. In the meantime, the tested sample solution added in the hole would completely spread around the hole to exert the antibacterial effect. If skip this step and put the plate directly into the incubator at 37 °C, the indicator bacterial would grow too quick to cover the whole plate and the antibacterial zone would be hard to observe.
Comment 6: “In materials and methods, section 2.3 lines 92-93, The three generation as they were determined? What mean bacterial generation?”
Response 6: The LAB used in this work were preserved in -80 °C ultra cold storage freezer in our lab. To resuscitate the strains, the colony of the LAB was inoculated into MRS broth and cultivated at 37 °C for 24 h. Then, the culture was transferred into fresh MRS broth in a ratio of 3% (v/v) and cultivated at 37 °C for 24 h, and repeat this process once. One generation means cultivation for 24 h.
Comment 7: “In materials and methods, section 2.6 lines 128-129, please describe more detailed the count of viable cells.”
Response 7: The count of viable cells has been modified as “The viable bacterial counts were performed as described [13]. Briefly, the sampled sulution was centrifuged (10000 rpm, 4 °C , 10 min) , obtaining the cell pellets. After washing twice by saline solution, the cell pellets were resuspended with the same volume of saline solution. One milliliter of the samples was diluted with 9 ml of saline solution, and eight serial dilutions were performed. Each bacterium was counted in the three most appropriate dilutions by applying the pour plate technique.”. These contents has been shown in the section “2.6. Viable count method in the revised manuscript.
Comment 8: “Section RESULTS, change for Results and discussion”
Response 8: The title has been revised as “Results and discussion” as your kind suggestion.
Comment 9: “In section of Results lines 151-161 is introduction, please eliminate this text.”
Response 9: Thank you for your recommendation. These content has been eliminated.
Comment 10: “In Results section, I suggest change the Table 1 for a figure with graphs or bars to show with clarity the results and a more detailed description of these results, for example which bacterial strains showed better antibacterial activity and selected.”
Response 10: We really appreciate your kind suggestion. As there are 102 of strains in Table 1, it is bigger than the magazine requirement after changing the table into a figure. To make it clear to understand, we have revised the contents in the corresponding sections of method and result.
Comment 11: “In Results section, Figure 1 and text (lines 165-173), not only the strain YT displayed antibacterial activity, also the isolate 58, Why it was discarded? One point important is that in figure 1 only the results for S. enterica is showed. Please edit the text.”
Response 11: The strain 58 cells only showed inhibitory activity to S. enterica and the corresponding inhibitory zone was smaller than that of strain YT cells (Table 3).
During the selection process, there are a lot of plates used to evaluate the inhibitory activity of the LAB to the indicator strains were . It is hard to show all of the plates in the manuscript. Therefore, some LAB strains were selected as representatives to be displayed in Figure 1. We are sorry we did not write this clearly in the manuscript. Therefore, the contents have been rewritten as “During the selection process, Therefore, the inhibitory activity to S. enterica of some LAB strains were used as representatives and showed in Figure 1.” in the revised manuscript.
Comment 12: “The word “blasted” (line 195) is incorrect, please edit.”
Response 12: The word “balsted” has been replaced by “aligned” in the revised manuscript.
Comment 13: “ In table 2, these values are in mm or what units? ”
Response 13: We are sorry we did not write clearly. The unit in table 2 has been revised as “ mm” in the revised manuscript.
Comment 14: “In Results section 3.3, figure 4, The cell growth of B.subtilis and S. enterica co-cultivated with Cs-YT, the results are confusing, because according with methodology is not clear as they were determined, also in these bacterial growth condition could be generate colonization competition, so that from my point of view these essays are not clear for interpretation.”
Response 14: The culture used in this section was LB broth in which the indicator strains were able to grow well while strain YT barely grow. Therefore, there were not colonization competition. Besides, the corresponding method to count viable bacteria has been supplied in the section 2.6 in the revised manuscript.
The manuscript have been thoroughly checked and modifications were made. Besides, Figure3, 4 , and 5 have been redrawn. We really hope the revised manuscript will be acceptable for publication.
Reviewer 3 Report
Comments and Suggestions for Authors
The topic of the manuscript: „Isolation and identification of Lacticaseibacillus rhamnosus YT and its antibacterial characteristics” falls within the thematic scope of FOODS journal.
The aim of the study was to isolate LAB of human origin with excellent ability to inhibit common foodborne pathogens and spoilage bacteria. General, research in the field of introduction to the development of new natural preservatives.
Below is a list of comments and observations - in the attached pdf file I have included all comments to the text, including those that I do not mention here because they are minor.
The manuscript requires corrections listed below:
1. Materials and methods - the methodology requires supplementing some information, the chapter does not describe the procedure clearly ("step by step"); discrepancies in information between the methodology and the description of results and conclusions (lack of information on the number of LAB strains tested at the beginning, then in Chapter 3 a different number than in Chapter 4 - Conclusions), lack of information on the methodology used for determining the number of the YT strain.
Line 122 - there is no information here on how the pathogen inoculum was prepared
2. Results – chapter 3 is called “3. Results” and there is no chapter “4. Discussion” - so maybe introduce “3. Results and discussion”?
a) Lines 165-174 - this is just a bit more of a description of the methodology, which is missing in section 2,
b) note to the entire manuscript - if we use the term lactobacilli (i.e. lactic bacilli) but not in the meaning of the genus name, we write it without italics and with a lowercase letter,
c) note to the entire manuscript - please always use the term "antibacterial activity" and do not introduce others (e.g. „antibacterial ability” or „capacity” etc. - see line 187),
d) note to the entire manuscript - several times in the manuscript (both in the methodology and in the description of results, on the figures) catalase or other enzymes are referred to as proteases - this must be changed,
e) note to Figures 2-5 - please indicate whether the figure shows average values ​​and whether the marked error bars represent standard deviation (another statistical parameter?).
All comments were introduced in the review mode to the attached pdf file.

Author Response
We appreciate your valuable comments on our manuscript. These comments and suggestions have been very carefully considered and the manuscript has been revised with modifications, as you kindly suggested.
Based on the order of the comments, the revised points are as listed below.
Comment 1: “Materials and methods - the methodology requires supplementing some information, the chapter does not describe the procedure clearly ("step by step"); discrepancies in information between the methodology and the description of results and conclusions (lack of information on the number of LAB strains tested at the beginning, then in Chapter 3 a different number than in Chapter 4 - Conclusions), lack of information on the methodology used for determining the number of the YT strain.”
Line 122 - there is no information here on how the pathogen inoculum was prepared
Respones 1:The manuscript has been thoroughly checked and the section “2. Materials and Methods” has been revised. The discrepancies and the lacked information that you kindly mentioned have been modified in the revised manuscript. Besides, the preparation of pathogen inoculum has been modified as “The indicator bacteria B. subtilis and S. enterica were seperately made by cultivation into a tube containing 5 mL of LB medium at 37 °C for 12 h. ” in the part of “2.7. Measurement of antibacterial ”
Comment 2: “ Results – chapter 3 is called “3. Results” and there is no chapter “4. Discussion” - so maybe introduce “3. Results and discussion”?”
Response 2: Thank you for your suggestion. “3. Results” has been revised as “3. Results and discussion”.
Comment 3 : “ Lines 165-174 - this is just a bit more of a description of the methodology, which is missing in section 2,”
Respones 3: This section and the corresponding method in section 2 have been modified in the revised manuscript.
Comment 4 : “ note to the entire manuscript - if we use the term lactobacilli (i.e. lactic bacilli) but not in the meaning of the genus name, we write it without italics and with a lowercase letter,”
Respones 4: Thank you for your suggestion. The manuscript has been thoroughly checked and 4 of Lactobacillus has been replaced by LAB.
Comment 5: “note to the entire manuscript - please always use the term "antibacterial activity" and do not introduce others (e.g. „antibacterial ability” or „capacity” etc. - see line 187),”
Respones 5: The term "antibacterial activity" has been unified in the revised manuscript.
Comment 6: “note to the entire manuscript - several times in the manuscript (both in the methodology and in the description of results, on the figures) catalase or other enzymes are referred to as proteases - this must be changed”
Respones 6: We are sorry for misunderstanding catalase as a kind of protease. The proteases including catalase has been changed in the revised manuscript.
Comment 7: “note to Figures 2-5 - please indicate whether the figure shows average values and whether the marked error bars represent standard deviation (another statistical parameter?).”
Respones 7: The data in the figures showed average values and the error bars represent standard deviation. The method of data analysis has been revised in the section “2.8. Statistical analysis”
Comment 8: All comments were introduced in the review mode to the attached pdf file.
Respones 8: The manuscript has been revised as your comments showed in pdf file.
The manuscript have been thoroughly checked and modifications were made. Besides, Figure3, 4 , and 5 have been redrawn. We really hope the revised manuscript will be acceptable for publication.
Round 2
Reviewer 1 Report
Comments and Suggestions for Authors
From my viewpoint, there are still bits and pieces related to the written english throughout the manuscript. I am sure the overall manuscript presentation will be better if you take into account such minor details.
Comments on the Quality of English LanguageThe written english has improved a lot compared to the prior version of the manuscript. However there are very minor details that the authors should be re-consider.
Reviewer 2 Report
Comments and Suggestions for Authors
The authors followed the comments suggested and the manuscript was sufficiently improved. I do not have more comments, therefore, it can be published in Foods.